# The Bile Acid-Phospholipid Conjugate Ursodeoxycholyl-Lysophosphatidylethanolamide (UDCA-LPE) Disintegrates the Lipid Backbone of Raft Plasma Membrane Domains by the Removal of the Membrane Phospholipase A2

**DOI:** 10.3390/ijms20225631

**Published:** 2019-11-11

**Authors:** Wolfgang Stremmel, Simone Staffer, Gert Fricker, Ralf Weiskirchen

**Affiliations:** 1Institute of Pharmacy and Molecular Biotechnology, University of Heidelberg, 69120 Heidelberg, Germany; gert.fricker@uni-hd.de; 2Department of Internal Medicine IV, University Hospital of Heidelberg, 69120 Heidelberg, Germany; simone.staffer@med.uni-heidelberg.de; 3Institute of Molecular Pathobiochemistry, Experimental Gene Therapy and Clinical Chemistry, RWTH University Hospital Aachen, 52074 Aachen, Germany; rweiskirchen@ukaachen.de

**Keywords:** liver, UDCA-PLE, NASH, fatty acids, inflammation, fibrosis, steatosis, therapy

## Abstract

The bile acid-phospholipid conjugate ursodeoxycholyl-lysophosphatidylethanolamide (UDCA-LPE) was shown to have anti-inflammatory, antisteatotic, and antifibrotic properties, rendering it as a drug targeting non-alcoholic steatohepatitis (NASH). On a molecular level, it disrupted the heterotetrameric fatty acid uptake complex localized in detergent-resistant membrane domains of the plasma membrane (DRM-PM). However, its mode of action was unclear. Methodologically, UDCA-LPE was incubated with the liver tumor cell line HepG2 as well as their isolated DRM-PM and all other cellular membranes (non-DRM). The membrane cholesterol and phospholipids were quantified as well as the DRM-PM protein composition by Western blotting. The results show a loss of DRM-PM by UDCA-LPE (50 µM) with a 63.13 ± 7.14% reduction of phospholipids and an 81.94 ± 8.30% reduction of cholesterol in relation to mg total protein. The ratio of phospholipids to cholesterol changed from 2:1 to 4:1, resembling those of non-DRM fractions. Among the members of the fatty acid uptake complex, the calcium-independent membrane phospholipase A2 (iPLA_2_β) abandoned DRM-PM most rapidly. As a consequence, the other members of this transport system disappeared as well as the DRM-PM anchored fibrosis regulating proteins integrin β-1 and lysophospholipid receptor 1 (LPAR-1). It is concluded that UDCA-LPE executes its action by iPLA_2_β removal from DRM-PM and consequent dissolution of the raft lipid platform.

## 1. Introduction

Detergent resistant membrane domains within the plasma membrane (DRM-PM) are of importance for cell signaling, metabolic control, and cell to cell communication. The lipid backbone of these microdomains consists of an ordered arrangement of phospholipids. Therein cholesterol fills the gap between the long fatty acid stretches of sphingomyelin and the surface of the bilayer allowing the structure to be tight and compact [1].

The bile acid-phospholipid conjugate ursodeoxycholyl-lysophosphatidylethanolamide (UDCA-LPE) inhibits in a dose-dependent fashion hepatocellular influx of fatty acids with an IC_50_ of 47 µM [2]. The positive effects on steatosis, inflammation, and fibrosis were only observed with this conjugate, but not with UDCA or LPE alone [3,4,5]. Mechanistically it was shown to remove proteins from DRM-PM, i.e., the members of the fatty acid uptake complex, which is formed by caveolin-1, the membrane fatty acid binding protein (FABP_PM_), the cluster of differentiation (CD36) and the calcium-independent membrane phospholipase A2 (iPLA_2_β) also known as PNPLA9 [2]. However, the mechanism how these proteins are removed from their lipid platform remains unclear. The question arises, whether the proteins are removed primarily or the structure of the DRM-PM, per se, is disrupted. The latter is supported by the detergent-like character of the bipolar UDCA-LPE molecule, where the lipophilic LPE moiety could anchor within the phospholipid bilayer and solubilize the lipid structure. Another consideration is an extraction of lipids, i.e., cholesterol, as it is proposed for cyclodextrin [6,7]. Accordingly, DRM-PM localized proteins lose their lipid backbone and are then removed. Alternatively, the DRM-PM proteins interact with UDCA-LPE and are thereof removed. As the particular target, iPLA_2_β can be identified. This enzyme appears only to be indirectly involved in the fatty acid influx process. It rather acts as a constitutive protein of the heterotetrameric uptake complex. In previous kinetic analyses, it was indeed suggested that UDCA-LPE binds to iPLA_2_β because it was shown to inhibit its enzymatic function in non-competitive fashion [2]. As a consequence of this structural interaction, iPLA_2_β could be removed together with bound phosphatidylcholine from the platform and the other proteins of the fatty acid uptake complex as well as other raft proteins stepwise fade from DRM-PM.

To address this problem, we analyzed the effect of UDCA-LPE on protein and lipid composition. For determination of the lipid dissolution of raft microdomains, we utilized the fact that plasma membranes in total contain a high proportion of cholesterol, reaching about 30% of the lipid bilayer [7]. Within those the DRM-PM microdomains are particularly rich in cholesterol, which in its unesterified form is abundant in membranes. Concerning DRM-PM phospholipids, predominant are phosphatidylcholine and sphingomyelin [1,7]. For the purpose of this study, we chose to determine total phospholipids and cholesterol and their ratio by a quantitative enzymatic assay to follow the fate of DRM-PMs. For analysis of time-dependent removal of raft localized proteins, isolated DRM-PM fractions were employed. They were prepared from the immortalized, human hepatocyte-derived tumor cell line HepG2 which in previous studies was shown to behave, in regard to fatty acid influx, like primary hepatocytes and to contain a raft localized fatty acid uptake complex [2].

## 2. Results

As expected, short term incubation of increasing doses of UDCA-LPE with HepG2 cells did not change the total phospholipid and cholesterol content in the homogenate. However, immunoprecipitation with antibodies to iPLA_2_β revealed in the homogenate after UDCA-LPE (0–100 µM) a dose-dependent decrease of cholesterol from 1.34 ± 0.10 to 0.23 ± 0.03 µmol × mg protein^−1^ (*p* < 0.001) with an IC_50_ at 31 µM. In this analysis, the phospholipid content remained unchanged.

The finding of unaltered phospholipid content in the iPLA_2_β immunoprecipitate of the homogenate is in contrast to immunoprecipitation with flotillin-1, which is a key protein to establish DRM-PM microdomains involved in endocytosis, signal transduction, and cytoskeleton regulation [8]. With flotillin-1 immunoprecipitation, phospholipids as well as cholesterol were reduced by 61.5% and 80.0%, respectively, after pretreatment of HepG2 cells with 50 µM UDCA-LPE (Figure 1).

The decrease in cholesterol was attributed to the DRM-PM fraction with an 81.94% ± 8.30% reduction per mg protein in the presence of 50 µM UDCA-LPE compared to controls (Figure 1). In these incubations with isolated DRM-PM also phospholipids were reduced by 63.13% ± 7.14%. In total, the ratio of phospholipids to cholesterol changed from 2:1 to 4:1 [7], which is a typical feature of non-DRM. The data indicated a loss of the DRM-PM fraction. UDCA-LPE did not affect the non-DRM fraction.

The unchanged phospholipid content after UDCA-LPE in the anti-iPLA_2_β immunoprecipitates of homogenates indicates that this enzyme binds phospholipids not only from DRM-PM, but also from other cell compartments. It was indeed shown previously that iPLA_2_β distributes to subcellular membranes other than DRM-PM and even cytosol from where it is immunoprecipitated with bound phospholipids [2]. The intrinsic phospholipid-binding capacity [9] is not shared by flotillin-1.

Thus, the DRM-PM lipid platform may be formed by phospholipid-binding proteins, such as iPLA_2_β. In addition, there are constitutive raft proteins, such as flotillin-1 which are not responsible for the characteristic lipid composition.

When the lipid platform building proteins are removed, the number of DRM-PMs is reduced. Remaining DRM-PMs still carry their constitutive proteins until the structural lipid backbone is dissolved. Therefore, we next tested the composition of DRM-PM proteins as a function of UDCA-LPE exposure over time (Figure 2).

iPLA_2_β and the associated members of the fatty acid uptake transporter complex, caveolin-1 and CD36 disappeared early. The proteins integrin β-1 (ITGB1) and lysophosphatidic acid receptor 1 (LPAR1), which are known actors in hepatic fibrogenesis, only gradually fade [2]. Flotillin-1 stays as longest watch-tower (Figure 2). However, with more time and higher concentrations, flotillin-1 also disappears.

## 3. Discussion

In previous studies it was shown that UDCA-LPE inhibits hepatocellular fatty acid influx by displacement of the heterotetrameric fatty acid uptake complex from DRM-PM. As control player iPLA_2_β was identified. However, it remained unclear whether fading of this enzyme was due to the detergent effect of UDCA-LPE or the UDCA-LPE coupled binding and consequent removal of iPLA_2_β. This is of mechanistical interest as well as of importance for development of a suitable therapeutic agent employing UDCA-LPE analogues. Although it is known that iPLA_2_β significantly contributes to important physiological processes, including inflammation, calcium homeostasis, and apoptosis, the underlying mechanisms of all these effects remain poorly understood [10]. However, the structure of iPLA_2_β that was recently solved suggests that this protein forms a stable dimer, in which the active sites of the dimer are wide open providing sufficient space for phospholipids to access the catalytic centers. The activity of this dimer can be allosterically inhibited by a single calmodulin altering the conformation of the dimerization interface [10]. It is obvious that such inhibitory binding partners or drugs interfering with the activity of iPLA_2_β are potentially relevant for treatment of inflammation.

Our experiments in HepG2 cells indicate that UDCA-LPE executes its biological activities by iPLA_2_β removal from DRM-PM and consequent dissolution of the raft lipid platform. Although these findings are very traceable and highly reproducible, a shortcoming of this study is the determination of total phospholipids and total cholesterol by an enzymatic assay. Preferably, mass spectrometry of all phospholipid species and cholesterol/cholesterol esters would have to be performed. However, the ratio of phospholipid to cholesterol gives insight into the distribution between DRM-PMs and other cell membranes (non-DRM) after UDCA-LPE exposures. Thus, the present work shows the dissolution of the DRM-PM lipid platform by exposure to UDCA-LPE. It occurs gradually and starts with the removal of iPLA_2_β with its bound phospholipids and consequent loss of cholesterol, both of which constitute the structural units of raft plasma membrane microdomains.

The iPLA_2_β represents the “bracket” protein of the heterotetrameric fatty acid uptake complex which upon UDCA-LPE loses contact to the DRM-PM platform and distributes to other cellular compartments [2]. As shown in previous experiments with isolated DRM-PMs, the process is rapid and already 30 min after UDCA-LPE exposure the removal is completed. The disappearance of the membrane fatty acid uptake complex is associated with diminished cellular influx of fatty acids which can be used as therapeutic strategy to fight steatosis as well as consequent inflammation and fibrosis [2,3,4,5,11,12,13]. Furthermore, the associated inhibition of iPLA_2_β suppresses the generation of lysophosphatidylcholine (LPC) from phosphatidylcholine (PC) [2,14]. As a consequence, the generation of phosphorylated c-Jun N-terminal kinase (pJNK-1) is inhibited, which is one of the key players in hepatic fibrogenesis [15].

The remaining quantitatively reduced number of raft microdomains still contains other constitutive proteins. As examples integrin β-1 and PLAR1 were particularly focused on, because they display an intracellular activation mechanism for fibrogenesis [2]. With a delay of 60 min UDCA-LPE exposure, they disappear from DRM-PM and prohibit downstream pathways essential for binding of integrins to the extracellular matrix [16]. Consequentially, recruitment and activation of essential signaling proteins for fibrogenesis are inhibited, such as focal adhesion kinase (FAK) and proto-oncogene tyrosine-protein kinase Src [17,18,19].

As last flag protein of DRM-PM, flotillin-1 disappears. It is involved in vesicular trafficking and signal transduction [20]. When it is removed from DRM-PM it indicates loss of this plasma membrane domain.

The mechanism behind the UDCA-LPE effect on DRM-PM remains speculative. If it is a pure detergent effect, one would expect the simultaneous disappearance of all raft plasma membrane proteins. This is not the case as we observed a rapid loss of the fatty acid uptake complex, a slow removal of integrins and a more prolonged stay of flotillin-1.

Therefore, it is a more appealing hypothesis when iPLA_2_β is primarily removed as key target of UDCA-LPE. Previous studies indeed showed that UDCA-LPE, a bile-acid-phospholipid conjugate, leads to a non-competitive inhibition of iPLA_2_β indicative for a conformational change of the enzyme which in addition may facilitate its removal from the fatty acid uptake complex as initial step. However, the direct demonstration of UDCA-LPE binding to iPLA_2_β within DRM-PM still represents a methodological challenge. From DRM-PMs, iPLA_2_β distributes to cytosol or other membrane compartments [2]. The iPLA_2_β is known to bind phospholipids [9] as this feature belongs to its genuine function as phospholipase. Accordingly, immunoprecipitation with anti-iPLA_2_β did not show a reduction of phospholipids in the homogenate but in the DRM-PM fraction. As a consequence of phospholipid removal from DRM-PM, cholesterol faded which is obvious after immunoprecipitation with representative raft proteins such as iPLA_2_β and flotillin-1. This was observed in the homogenate and DRM-PM fractions because cholesterol has no binding affinity to these proteins. Most likely cholesterol distributes to cytosol which was not further investigated in this study.

The positive effects of UDCA-LPE on steatosis [2,11], hepatic regeneration [4], inflammation [4,5,12], and in particular on fibrosis [3,13] can now mechanistically better be explained: It is the dissolution of DRM-PM as platforms of the raft plasma membrane microdomains which is initiated by the removal of iPLA_2_β. One consequence is the inhibition of hepatic fibrosis which was shown to be triggered via suppression of DRM-PM dependent pJNK-1 and integrin activation [2,3]. The rigid structure of these specialized lipid membrane domains can easily be measured by a membrane fluidity analysis. Reduced membrane fluidity due to more DRM-PMs indicated imminent fibrosis development whereas enhanced fluidity with less DRM-PMs is considered protective [21]. Such a protective increase of membrane fluidity was suggested for a long time and can now be substantiated by an underlying mechanism. However, further experimental prove is required.

If UDCA-LPE acts as iPLA_2_β inhibitor with the consequence of an anti-inflammatory and antifibrotic effect, it could be an interesting therapeutic tool. Initial animal experiments showed indeed promising results [3,13]. Until now no adverse events were observed in animals [3,4,5,11,12,13]. However, it needs attention whether loss of DRM-PM has negative biological consequences.

## 4. Materials and Methods

### 4.1. Tissue Culture Models

As previously described, HepG2 cells were grown to confluence for 16 h in Dulbecco’s minimal essential medium (MEM) containing 10% fetal calf serum (all reagents from Life Technologies, Thermo Fisher Scientific, Darmstadt, Germany) [2].

### 4.2. Isolation of Detergent-Resistant Membranes and Non-DRM Fractions

Detergent extraction was performed with 3-[3-cholamidopropyl) dimethylammonio]-1-propanesulfonate (CHAPS) [2]. In brief, HepG2 cells were rinsed with ice-cold phosphate-buffered saline (PBS) and scraped into ice cold 300 µL of 25 mM Tris-HCI (pH 7.4), 150 mM NaCl, 3 mM EDTA (TNE) buffer containing leupeptin, pepstatin, chymostatin, and antipain (each at 25 µg/mL). Cells were homogenized 15-times through a 22-gauge needle followed by 10 strokes with a tight-fitting Dounce homogenizer.

The lysate was centrifuged for 5 min at 3000 rpm to obtain a postnuclear supernatant, which was subjected to extraction with 20 mM CHAPS in TNE buffer on ice. The extracts were adjusted to 40% sucrose and overlaid with a discontinuous sucrose gradient (6 mL of 30% sucrose in TNE or 2 mL of TNE without sucrose). The gradients were centrifuged at 200,000× *g* in a Beckman SW41 rotor for 16–22 h at 4 °C. Supernatant DRM-PM fractions and non-DRM pellets were obtained and used for further studies [2,3].

### 4.3. Incubation Experiments

HepG2 cells were incubated with 1 mL 0–100 µM UDCA-LPE in PBS with 1 mM taurocholate at 37 °C for 60 min. Homogenates (10 mg/mL protein) were taken as such or after immunoprecipitation. DRM-PM or non-DRM (10 mg/mL protein) were incubated with UDCA-LPE or PBS in 1 mM taurocholate at 37 °C for 60 min. After centrifugation for 1 h at 100,000× *g*, the lipids were determined in pellets (phospholipid assay kit, cat. no.: KA1635, Abnova, Heidelberg, Germany, and cholesterol fluorometric assay kit, 10007640, Cayman-Chem, Ann Arbor, MI, USA) as well as pellets were used for Western blotting. The protein concentrations were determined using the colorimetric Bradford assay obtained from Bio-Rad, Munich, Germany.

### 4.4. Western Blotting

Immunoblotting was performed according to a standard protocol [2]. The following primary antibodies to human proteins were used: mouse α-CD36 (sc-70642; Santa Cruz Biotechnology Inc., Santa Cruz, CA, USA) at 1:200; rabbit α-iPLA_2_β (ab103258; Abcam, Hamburg, Germany) at 1:500; mouse α-FABP_PM_ (MAB10669; Abnova, Heidelberg, Germany) at 1:1000; mouse α-caveolin-1 (sc-135860; Santa Cruz) at 1:500; mouse α-flotillin-1 (sc-74567; Santa Cruz) at 1:500; mouse α-LPAR1 (H00001902-M08-100; Abnova) at 1:1000; goat α-integrin β-1 (sc-6622; Santa Cruz) at 1:1000; and mouse α-β-actin (A1978; Sigma-Aldrich, Taufkirchen, Germany) at 1:100,000. Anti-mouse and anti-rabbit horseradish peroxidase coupled antibodies (Dianova, Hamburg, Germany) were used at a 1:10,000 dilution as secondary antibodies.

### 4.5. Immunoprecipitation

Aliquots of homogenate samples (20 µg of protein in 80 µL) were added to 20 µL of solubilization buffer containing 0.25 mM 2-[N-morpholino]ethanesulfonic acid (MES) buffer (pH 6.5) containing 0.15 M NaCl, 0.05% Triton X-100 (T9284; Sigma-Aldrich), and a protein inhibitor cocktail (P8340; Sigma-Aldrich) as described before [2]. Samples were then incubated for 16 h at 4 °C with antibodies to iPLA_2_β or flotillin-1 at a 1:100 dilution; 20 µL of protein A/G Plus agarose beads (sc-2003; Santa Cruz) were then added, vortexed, and incubation resumed for 16 h at 4 °C with rotation. After centrifugation for 15 min at 2000 rpm pellets were washed twice in MES buffer and used for Western blotting as well as lipid quantification calculated to the amount of protein incubated.

### 4.6. Statistical Analyses

Reproducibility of each experiment was obtained from at least three independent experiments. All results were analyzed with Prism software version 5.0 (GraphPad Software, Inc., La Jolla, CA, USA). Data are presented as mean ± SD. *p* < 0.05 was considered significant by using pairwise Student *t*-tests, and one-way ANOVA followed by multiple comparison using Dunnett’s tests.

## 5. Conclusions

We conclude that UDCA-LPE executes its biological activities by iPLA2β removal from DRM-PM, thereby mediating anti-steatotic, -inflammatory and –fibrotic activities. In future, it is mandatory to test whether UDCA-LPE analogues with better pharmacokinetics and bioavailability as well as higher efficacy can become available for human trials.

## Figures and Tables

**Figure 1 ijms-20-05631-f001:**
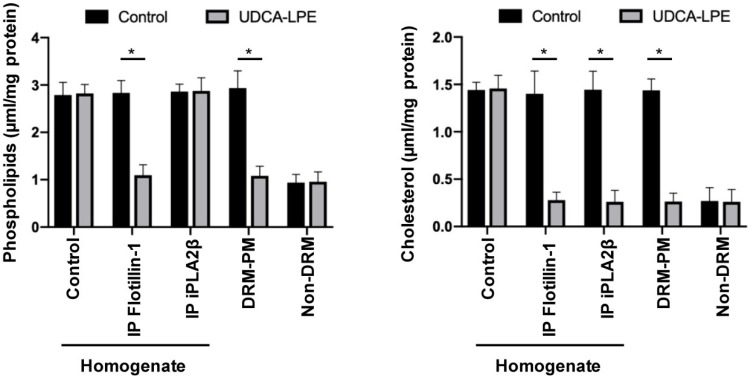
Lipid distribution as function of ursodeoxycholyl-lysophosphatidylethanolamide (UDCA-LPE) exposure. HepG2 cells were incubated for 60 min with 50 µM UDCA-LPE or as controls with phosphate-buffered saline (PBS). Samples with 10 mg/mL protein of HepG2 homogenate were taken as such or immunoprecipitated with antibodies directed against flotillin-1 or calcium-independent phospholipase A2 (iPLA_2_β). In comparison isolated detergent resistant membrane domains within the plasma membranes (DRM-PMs) and non-DRMs (10 mg/mL) were treated for 30 min with 50 µM UDCA-LPE or PBS as controls. After centrifugation for 100,000× *g* for 1 h, the pellets were resuspended, and lipids were determined and correlated to the initially applied protein concentration. Illustrated are means ± standard derivation of three repetitive experiments, * = *p* < 0.001.

**Figure 2 ijms-20-05631-f002:**
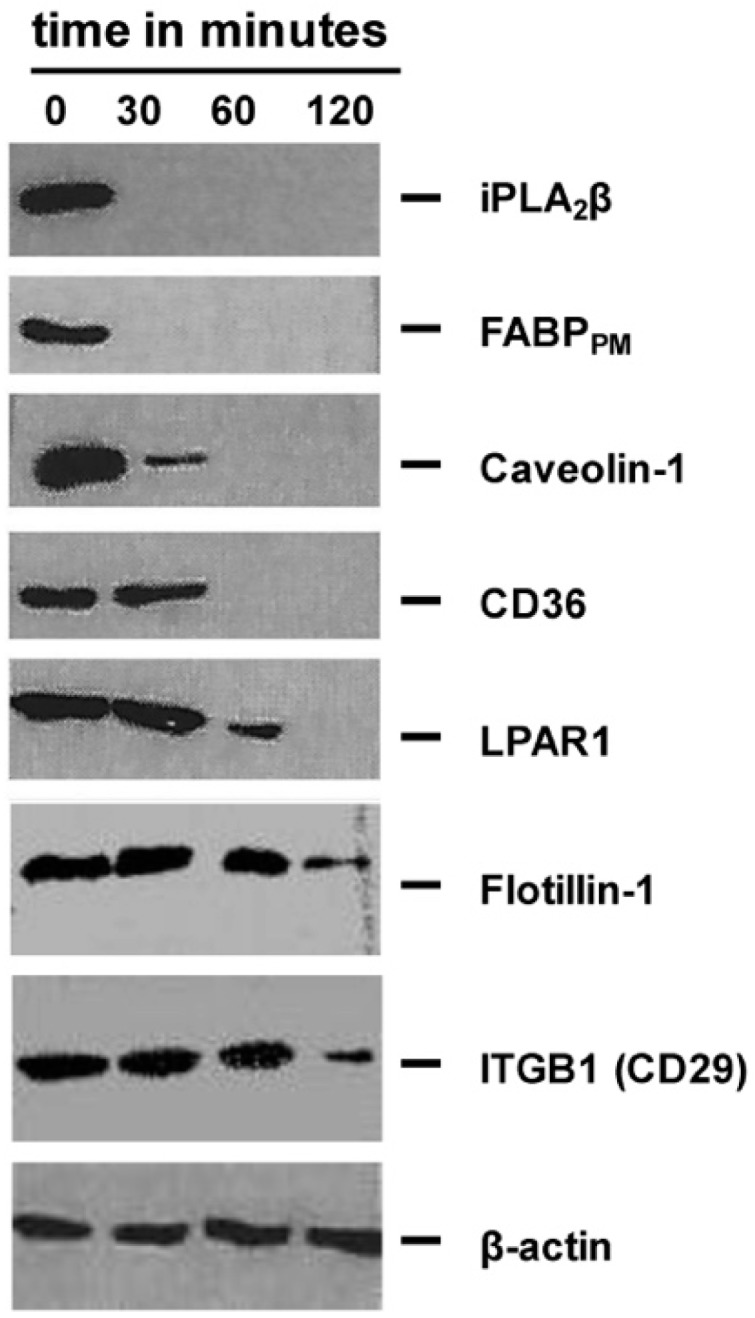
DRM-PM protein composition as function of UDCA-LPE exposure over time. Isolated native DRM-PMs (10 mg/mL) were incubated over a 120 min time frame with UDCA-LPE (50 µM). After incubation and centrifugation at 100,000 g for 1 h, Western blot of indicated proteins were performed and compared to β-actin as a loading control. Abbreviations used are: iPLA2β, calcium-independent membrane phospholipase A2; FABP_PM_, membrane fatty acid binding protein; CD36, cluster of differentiation 36; LPAR1, lysophosphatidic acid receptor 1; ITGB1, integrin β-1.

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
