# Peer review of "The Bile Acid-Phospholipid Conjugate Ursodeoxycholyl-Lysophosphatidylethanolamide (UDCA-LPE) Disintegrates the Lipid Backbone of Raft Plasma Membrane Domains by the Removal of the Membrane Phospholipase A2"

_ijms, 2019, doi:10.3390/ijms20225631_

Round 1

Reviewer 1 Report

In this study, the authors examined the potential mechanism of removal of heterotetrameric FA uptake protein complex using the in vitro model by treating HepG2 cells with UDCA-LPE. The presented data suggest that UDCA-LPE sequentially removed individual components of the FA complex from DRM-PM and then dissolve the lipid raft platform. This is an interesting study providing new information about FA uptake complex removal from the DRM-PM. This is basically an extension study from the author’s previous reports on the UDCA-LPEs application in the NASH pathological features

 Minor comments-

1.    One limitation of this study is the use of HepG2, which is a hepatic tumor cell line. The hepatic tumor cell line could be potentially different in terms of number and quality of lipid raft as compared to regular (normal) hepatocyte. This should be discussed to some extent.

2.    The presented title should be rephrased especially in terms of antifibrotic therapy as there is no data presented in relation to the antifibrosis process. It’s an over-interpretation of the data.

3.    Figure 1: the asterisks are misaligned.

Author Response

Reviewer 1

In this study, the authors examined the potential mechanism of removal of heterotetrameric FA uptake protein complex using the in vitro model by treating HepG2 cells with UDCA-LPE. The presented data suggest that UDCA-LPE sequentially removed individual components of the FA complex from DRM-PM and then dissolve the lipid raft platform. This is an interesting study providing new information about FA uptake complex removal from the DRM-PM. This is basically an extension study from the author’s previous reports on the UDCA-LPEs application in the NASH pathological features.

Dear reviewer,
Many thanks for taking the time to read our paper and the overall constructive comments.

Minor comments:
1. One limitation of this study is the use of HepG2, which is a hepatic tumor cell line. The hepatic tumor cell line could be potentially different in terms of number and quality of lipid raft as compared to regular (normal) hepatocyte. This should be discussed to some extent.

Many thanks for this comment. This is commented in line 69-70. We refer to the
characteristics of HepG2 cells used in a similar setting in a previous publication.

2. The presented title should be rephrased especially in terms of antifibrotic therapy as there is no data presented in relation to the antifibrosis process. It’s an over-interpretation of the data.

We have rephrased the title, not focusing on the antifibrotic effect.

3. Figure 1: the asterisks are misaligned.

Thank you very much for noticing it. We have replaced the asterisks in Figure 1.

Reviewer 2 Report

In this study, Stremmel et al. seek effects of UDCA-LPE on DRM-PM. The authors have published a previous paper (ref.#2), and this previous study covers almost everything shown in this manuscript. This manuscript and data look like supplementary figures for ref.#2 and I cannot find any novel findings that worth for publication in this manuscript.

Author Response

Reviewer 2

In this study, Stremmel et al. seek effects of UDCA-LPE on DRM-PM. The authors have published a previous paper (ref.#2), and this previous study covers almost everything shown in this manuscript. This manuscript and data look like supplementary figures for ref.#2 and I cannot find any novel findings that worth for publication in this manuscript.

Dear reviewer,
The novelty of the present study is the analysis of the phospholipid and cholesterol content in relation to the protein composition of DRM-PM. In the present study isolated DRM-PM were analyzed.
This is now mentioned in line 59-60 as well as in the last paragraph of the introduction.

Reviewer 3 Report

The manuscript ”The Bile Acid-Phospholipid Conjugate Ursodeoxycholyl-lysophosphatidylethanolamide (UDCA-LPE) Disintegrates The Lipid Backbone of Raft Plasma Membrane Domains: Implication for An Antifibrotic Therapy” by Stremmel et al. shows that UDCA-LPE causes depletion of phospholipids and cholesterol in DRM-PM of HepG2 cells. This causes the dissociation of the fatty acid uptake complex.

Abstract: “ As a consequence, the fibrosis regulating DRM-PM proteins integrin β-1 and lysophospholipid receptor 1 (LPAR-1) disappeared from DRM-PM. It is concluded that UDCA-LPE executes its antifibrotic action by iPLA2β removal from DRM-PM and consequent dissolution of the raft lipid platform.” There are no data shown regarding integrins and thus it is not appropriate to refer to it in the Abstract.

Introduction, please explain composition and function of non-DRM-PM. Flotillin-1 should be also shortly described.

”the remaining dominant phospholipid is sphingomyelin” please correct

The liver has a high content of free cholesterol. Effect of UDCA-LPE on the abundance of free cholesterol and cholesteryl ester should be analysed.

Subcellular fractionation refers to reference 2, here it is explained: “Detergent-resistant plasma membrane (DRM-PM) and non-DRM-PM fractions were isolated and characterized as described previously (19). Subcellular membrane fractions were prepared by centrifugation at 100,000 g for 1 h to obtain subcellular membranes along with cytosolic supernatants.” This method is important for the paper and should be described in more detail.

Figure 2, data should be quantified and shown in a graph.

“with more time and higher concentrations, Flotillin-1 also disappears.” Authors may show these data.

“therapeutic strategy to fight steatosis“ what are the benefits when steatosis improves? To what extend contributes fatty acid uptake to liver steatosis?

Author Response

Reviewer 3

The manuscript ”The Bile Acid-Phospholipid Conjugate Ursodeoxycholyllysophosphatidylethanolamide (UDCA-LPE) Disintegrates The Lipid Backbone of Raft Plasma Membrane Domains: Implication for An Antifibrotic Therapy” by Stremmel et al. shows that UDCA-LPE causes depletion of phospholipids and cholesterol in DRM-PM of HepG2 cells. This causes the dissociation of the fatty acid uptake complex.

Dear reviewer,
Many thanks for taking the time to read our paper and the overall constructive
comments.

Abstract: “As a consequence, the fibrosis regulating DRM-PM proteins integrin β-1 and lysophospholipid receptor 1 (LPAR-1) disappeared from DRM-PM. It is concluded that UDCA-LPE executes its antifibrotic action by iPLA2β removal from DRM-PM and consequent dissolution of the raft lipid platform.” There are no data shown regarding integrins and thus it is not appropriate to refer to it in the Abstract.

Data of integrin-β1 are now included in Figure 2.The data are also discussed in lines 211-217.

Introduction, please explain composition and function of non-DRM-PM. Flotillin-1
should be also shortly described.

Non-DRM-PM has been explained and changed to non-DRM (s. abstract) and throughout the text accordingly. Flotilin-1 is a key protein to establish microdomains in the plasma membrane. They function in processes of endocytosis, signal transduction and regulation of cytoskeleton (8). This is now included on lines 132-134.

”the remaining dominant phospholipid is sphingomyelin” please correct

This sentence is now rephrased (please see lines 63-64).

The liver has a high content of free cholesterol. Effect of UDCA-LPE on the
abundance of free cholesterol and cholesteryl ester should be analysed.

We determined total cholesterol only, because it is the free cholesterol, which is of structural relevance for DRM-PM constitution (1,7). Cholesterol esters are the preferred form of cholesterol for transport in plasma and storage. For purpose of simplicity and available resources employed in this study, we chose to determine
total phospholipids and total cholesterol only. This is mentioned in lines 62-66. The shortcoming of this strategy is now discussed in lines 195-197.

Subcellular fractionation refers to reference 2, here it is explained: “Detergentresistant plasma membrane (DRM-PM) and non-DRM-PM fractions were isolated and characterized as described previously (19). Subcellular membrane fractions were prepared by centrifugation at 100,000 g for 1 h to obtain subcellular membranes along with cytosolic supernatants.” This method is important for the paper and should be described in more detail.

As suggested, we have added more details describing the subcellular fractionation under 2.2.

Figure 2, data should be quantified and shown in a graph.

We chose illustration of a representative Western blot instead of a quantitative
evaluation prone to artifacts.

“with more time and higher concentrations, Flotillin-1 also disappears.” Authors may show these data.

The disappearance of Flotillin-1 is now shown in Figure 2.

“therapeutic strategy to fight steatosis“ what are the benefits when steatosis
improves? To what extend contributes fatty acid uptake to liver steatosis?

This is indeed an interesting question. If reduced uptake and unchanged excretion occurs, there is a net reduction of fat within cells. This was previously shown by treating HepG2 cells with UDCA-LPE (2). This led to reduction of inflammation (LDH elevation) (2).This was also demonstrated in vivo (11).In addition, it was shown that fibrosis was suppressed (3,13). All of this is of benefit for patients with steatohepatitis. This is now included in lines 203-205.

Reviewer 4 Report

The following is a review of a paper on how UDCA-LPE disintegrates the lipid backbone of raft plasma membranes by Stremmel and colleauges. This selectively inhibits iPLA2B which may partially explain therapeutic effects of UDCA-LPE. The manuscript is well written and appears to be well performed but is fairly limited in overall data in numerous regards.  Some simple experiments could be added to improve the quality of the manuscript.

My primary concern is that the manuscript is done with one dose of UDCA-LPE in one cell line. The authors try to extrapolate this very isolated piece of data to liver fibrosis which is a multi-cellular complexity present in advanced liver disease. It would help validate these experiments if the authors demonstrated dose responsiveness of the effect, or at least multiple doses. An additional major concern is the use of HepG2 cells which are a hepatoblastoma-like cancer line. These cells do not express NTPC, the major bile acid transporter. Because of this, they may accumulate more UDCA-LPE than normal hepatocytes would as UDCA is minimally actively transported across the plasma membrane. Since the proposed therapeutic use of UDCA-LPE is in normal cells, the authors should repeat these data in isolated mouse/rat hepatocytes or at the very least in another liver cell line. I would be very interested to know if this effect is primarily due to UDCA, LPE, or is a real effect of the conjugate. It seems like this would be a critical control, as UDCA itself is highly bioactive and is used as a frontline therapeutic in PSC/PBC.

Author Response

Reviewer 4

The following is a review of a paper on how UDCA-LPE disintegrates the lipid
backbone of raft plasma membranes by Stremmel and colleauges. This selectively inhibits iPLA2B which may partially explain therapeutic effects of UDCA-LPE. The manuscript is well written and appears to be well performed but is fairly limited in overall data in numerous regards. Some simple experiments could be added to improve the quality of the manuscript.

Dear reviewer,
Many thanks for taking the time to read our paper and the overall constructive
comments.

My primary concern is that the manuscript is done with one dose of UDCA-LPE in
one cell line. The authors try to extrapolate this very isolated piece of data to liver fibrosis which is a multi-cellular complexity present in advanced liver disease. It would help validate these experiments if the authors demonstrated dose responsiveness of the effect, or at least multiple doses.

The dose responsiveness of UDCA-LPE on raft dissolution was shown before in vitro (2). The inhibition of fatty acid influx by UDCA-LPE as optimal quantitative parameter revealed an IC50 of 47µm (2). This is now stated on lines 38-40.

An additional major concern is the use of HepG2 cells which are a hepatoblastomalike cancer line. These cells do not express NTPC, the major bile acid transporter. Because of this, they may accumulate more UDCA-LPE than normal hepatocytes would as UDCA is minimally actively transported across the plasma membrane. Since the proposed therapeutic use of UDCA-LPE is in normal cells, the authors should repeat these data in isolated mouse/rat hepatocytes or at the very least in another liver cell line.

The potential shortcoming of use of HepG2 cells is appreciated and mentioned now in lines 67-70. We refer to our previous paper (2).

I would be very interested to know if this effect is primarily due to UDCA, LPE, or is a real effect of the conjugate. It seems like this would be a critical control, as UDCA itself is highly bioactive and is used as a frontline therapeutic in PSC/PBC.

The different effects of UDCA, LPE and the conjugate UDCA-LPE were examined
previously and it was demonstrated that only UDCA-LPE is effective in suppressing steatosis , inflammation and fibrosis (3-5). This is now stated on line 40-41.

Round 2

Reviewer 2 Report

I do not have comments.

Reviewer 4 Report

the authors improvements satisfy my concerns with the paper.